# Polymer Chemistry Defines Adjuvant Properties and Determines the Immune Response against the Antigen or Vaccine

**DOI:** 10.3390/vaccines11091395

**Published:** 2023-08-22

**Authors:** Akhilesh Kumar Shakya, Kutty Selva Nandakumar

**Affiliations:** 1Whitacre College of Engineering, Texas Tech University, Lubbock, TX 79409, USA; 2Department of Environmental and Biosciences, School of Business, Innovation and Sustainability, Halmstad University, 30118 Halmstad, Sweden

**Keywords:** immune response, polymer chemistry, antigen, drug delivery, functional moiety, cancer, autoimmunity, polymer

## Abstract

Activation of the immune system is a needed for designing new antigen/drug delivery systems to develop new therapeutics and for developing animal disease models to study the disease pathogenesis. A weak antigen alone is insufficient to activate the immune system. Sometimes, assistance in the form of polymers is needed to control the release of antigens under in vivo conditions or in the form of an adjuvant to activate the immune system efficiently. Many kinds of polymers from different functional groups are suitable as microbial antigens for inducing therapeutic immune responses against infectious diseases at the preclinical level. The choice of the functionality of polymer varies as per the application type. Polymers from the acid and ester groups are the most common types investigated for protein-based antigens. However, electrostatic interaction-displaying polymers like cationic polymers are the most common type for nucleic acid-based antigens. Metal coordination chemistry is commonly used in polymers designed for cancer immunotherapeutic applications to suppress inflammation and induce a protective immune response. Amide chemistry is widely deployed in polymers used to develop antigen-specific disease models like the experimental autoimmune arthritis murine model.

## 1. Introduction

Understanding the influence of polymers on the immune system is an essential prerequisite to applying various polymers for biomedical applications. Over the past several decades, immunological research has shifted towards polymer science to design a biocompatible system for delivering antigens/vaccines and activating the immune system [1]. The use of polymers gained popularity in immunological applications due to their ease of synthesis, characterization, and customization according to the nature of the antigen. Polymers physically or chemically hold the antigen and release it slowly, thus acting as a depot generation to keep activating the immune system for a long duration [1,2].

The polymers used in immunological applications are biocompatible and biodegradable while still assisting in activation of the immune system via regulating the release of an antigen. Polymers also offer the advantage of easy conjugation of biological moieties such as antigens/ligands to enhance the specificity of a delivery system. Atom transfer radical polymerization and reversible addition–fragmentation chain transfer polymerization methods facilitated polymer synthesis by adding many commercially available and custom-made monomers with specific functionality and unique polymer structures [3,4]. Interestingly, functional properties from several monomers can be incorporated into a single macromolecule using the reversible-deactivation radical polymerization (RDRP) procedure.

The immune response mediated by polymer and antigen generally depends on extrinsic and intrinsic properties like physical format, molecular weight, and nature of monomers involved in the polymer [2]. Based on the nature of the antigen, an appropriate balance of extrinsic and intrinsic properties is needed to design an effective way to activate the immune system. From an immunological point of view, polymers act as an adjuvant to improve the efficacy of an antigen [5,6,7,8]. Generally, an adjuvant helps antigens presented to the antigen-presenting cells (APCs) and enhances the costimulatory signals for activation of Th cells. Activated Th cells downstream activate the plasma B cells to produce the antibodies against the delivered antigens [1,2]. Although many specific cell signaling pathways are involved in polymer-mediated activation of the immune system, the polymers can act as agonists to different pattern recognition receptors (PRRs) [9]. Injected polymers are recognized by various PRRs present in the host cells, including, for example, integrins like αMβ2 (Mac-1); toll-like receptors and C-type lectin receptors involved in the regulation of inflammation and the activation of immune responses; scavenger receptors like SR-A I, SR-A II, and MARCO; and other surface proteins interacting with any foreign body [10]. Apart from recognition by PRRs, depot generation and the activation of complement pathways by polymers are likely to play a significant role in activating the immune system [11,12]. When polymers are introduced into the host, a protein layer is adsorbed onto the polymer surface [13], which could mediate the immune responses [14,15]. Pre-adsorption of fibrinogen and other serum proteins like antibodies, complement component C3, fibronectin, high molecular weight kininogen, vitronectin, and coagulation cascade proteins, especially on the hydrophobic polymer surface, could very well initiate the host response to such polymers, leading to typical inflammatory responses [16,17]. Protein coating on the polymers attracts the inflammatory cells, like monocytes, macrophages, and neutrophils, that can amplify the inflammatory cascade by secreting various oxygen and nitrogen radicals and granules [18] apart from proteolytic enzymes, cytokines, and chemokines. Size, shape, surface charge, hydrophobicity, controlled release of antigens, encapsulation of co-adjuvants, and the capacity to induce cell- and humoral-mediated immune responses of polymers contributing to their adjuvant properties were discussed in detail elsewhere [19].

Fine tuning of polymer can directly or indirectly affect the antigen-specific immune response. The parameters including chemistry of monomers, molecular weight, and format have been found to be important factors in defining the polymer-mediated antigen immune response. The effect of molecular weight and formats were discussed previously [20,21]. This review is primarily focused on the chemistry of polymers and their effects on the immune system (Table 1).

The following paragraphs discuss different polymer chemistries and their role in deciphering antigen-specific immune response.

## 2. Anhydride Chemistry

Polyanhydrides are the class of biodegradable polymers which are the favorite choice for use as a carrier system to deliver cancer vaccines. Polyanhydrides are biodegradable and FDA approved. Polyanhydrides follow zero-order release kinetics, using surface erosion to release the antigen over a longer duration [32]. A slow and long duration release of an antigen can be helpful to maintain immunity, especially in cancer treatment [33]. The antigen/protein release profile can be tuned easily according to the application via changing the monomer type and its ratio in copolymer composition. The balance of both charges in the form of amphiphilic copolymers represents a promising system to maintain the functionality of encapsulated biomolecules and their release in a specific environment. In the context of immune system activation, polyanhydrides have been demonstrated to be an adjuvant in innate immunity via acting as possible agonists for toll-like receptors [34].

To study the effect of polyanhydride chemistry (PAC), polyanhydride-containing copolymers sebacic anhydride (SA), 1,8-bis-(p-carboxy phenoxy)-3,6-dioxaoctane (CPTEG), and 1,6-bis-(p-carboxy phenoxy)-hexane (CPH) in different combination-based nanoparticles were studied with Ova as a model antigen (Figure 1). Three different compositions of CPTEG+CPH (50:50), CPTEG+CPH (20:80), and CPH+SA (20:80) NPs encapsulated Ova with or without CpG as an adjuvant were evaluated in the prevention of cancer development in C57BL/6J mice. The NPs were synthesized through a double emulsion solvent evaporation method with a comparable size range regardless of the combination. An unprotonated carboxylic acid may account for the negative charge, which might facilitate the attachment of Ova on the NP’s surface. Increased negative charges on the polymer increase the hydrophilicity, which produces a high encapsulation efficiency. The high hydrophilic content in this polymer is SA, which affects the Ovalbumin encapsulation efficiency in these polymeric NPs in the following order: 20:80 CPH+SA > 50:50 CPTEG+CPH > 20:80 CPTEG+CPH. The surface chemistry of the NPs influences their interaction with the antigen-presenting cells, which, finally, generates the magnitude of the immune response. The high hydrophobicity of particles promotes a greater opsonization of particles, which may be attributed to the danger signals that can facilitate and generate the immune response. Moreover, hydrophobicity releases the antigen slowly and continuously, which might enhance the sustaining of an ongoing immune response. A similar effect of hydrophobicity was observed in PAC-containing NPs (20:80 CPTEG+CPH). In mice, the 20:80 CPTEG+CPH NPs generated the highest level of Ova-specific antibodies and CD8+ T lymphocytes compared to the other compositions’ NP formulation. Moreover, in mice immunized with two subcutaneous injections of 20:80 CPTEG+CPH NPs encapsulated with Ova, a greater protection against tumor development was induced than with the other combination [22]. Overall, polyanhydrides have been proven as an effective adjuvant to sustain the release of a model or different cancer antigens. However, their role in controlling microbial antigens needs to be addressed in the future.

## 3. Zwitterionic Chemistry

Generally, zwitterion polymers possess equal parts of cationic and anionic components to maintain a neutral charge. The combination of both charges gives them good hydration properties. Because of this hydration property, a zwitterionic polymer can inhibit the nonspecific interaction of proteins [35]. Thus, this unique property makes them attractive to biomedical scientists, especially for bioengineering applications [36]. However, for the first time, their immunomodulation property was demonstrated by Li and colleagues, who synthesized a zwitterionic phosphoserine-mimetic polymer (ZPS)–uricase conjugate. The enzyme conjugates inhibiting the immunogenicity properties of the enzyme showed an improvement in their pharmacokinetic profile. Under in-vitro conditions, the enzyme conjugate considerably affected the antigen-presenting cells by suppressing the expression of cell surface maturation markers CD40^+^CD80^+^ compared to the naïve enzyme [37]. Another zwitterionic polymer, poly(carboxybetaine methacrylate) (pCBMA), which exhibited the nonspecific protein interaction, has shown promising application in the formation of complex blood media. Structurally, CBMA contains a glycine and betaine structure and exhibits an acid–base equilibrium. Carboxyl groups of CBMA are available to conjugate for amino groups on the protein surface [38]. A microgel based on pCBMA cross-linked with tetratheylene glycol diacrylate (TTEGDA) has demonstrated effectiveness for immunoglobulin (Ig) delivery. The microgel has shown good biocompatibility and sustains released immunoglobulin (Ig) under in-vitro conditions. The release Ig was functional and showed binding ability to Ig receptors [23].

Qiao and colleagues have modified cationic liposomes with zwitterionic lipids (distearoyl phosphoethanolamine-polycarboxylic- betaine, DSPE-PCB) to facilitate cellular uptake of DNA vaccines and increased release of DNA. In their study, they used zwitterionic lipid mannosylated DSPE-PCB (mannose-DSPE-PCB), cationic lipid, 1,2-Dioleoyl-3-trimethylammonium propane (DOTAP), and cholesterol as the helper lipid to develop a DNA adjuvant. In this system, mannose-DSPE-PCB enhanced DNA’s cellular accumulation and antigen presentation, DOTAP helped to complex with the DNA, and cholesterol stabilized the cationic liposomes. HIV DNA plasmid Env was used as the antigen. These man-ZCL/DNA lipoplexes have significantly increased the immunogenicity and anti-HIV immune responses compared to naked DNA, CpG/DNA, and Lipo2k/DNA [39].

Moreover, zwitterionic bacterial polysaccharides possessing highly dense positive- and negative-charged carbohydrate residues can be used as adjuvants [40]. The structure of natural zwitterionic polysaccharides and their mode of action, especially in activating acquired and innate immune responses, were discussed earlier [41,42]. Natural polysaccharides from Group B *Streptococcus* bacteria capsules are anionic and T-independent antigens. Zwitterionic polysaccharides (ZPS) synthesized through the chemical introduction of positive charges into these anionic polysaccharides activated human and mouse antigen-presenting cells through toll-like receptor 2 [43]. Later, the authors demonstrated that ZPS-containing glycoconjugate vaccines were more immunogenic than the native polysaccharide conjugates [44]. Despite their effectiveness, comprehensive biocompatibility characterization is needed to further explore zwitterionic polymers for immunological applications.

## 4. Amide Chemistry

Amide chemistry has biological importance, as it combines both hydrophilicity and hydrophobicity properties. Structurally, the primary amine group of amide provides an essential character, while the carboxyl group contributes to its acidic properties. A familiar example of amide chemistry’s role in activating the immune system was observed in the studies of a synthetic poly-*N*-isopropyl acrylamide polymer (PNiPAAm) as an adjuvant. Subcutaneous injection of PNiPAAm mixed with collagen type II (CII) induced a considerably higher CII-specific immune response than the antigen alone. Molecular weight also affects the adjuvant property of PNiPAAm, as high Mw PNiPAAm generated a higher anti-CII response than low Mw PNiPAAm. Moreover, the physical interaction of CII looks more promising than the CII covalently attached to PNiPAAm for activation of CII-specific immune responses [6,7]. PNiPAAm behaved differently at different temperatures, thus making it an essential component for the delivery of biomolecules. A temperature point widely known as a cloud point or lower critical solution temperature below it, PNiPAAm, remains in a solution form that allows the incorporation of biomolecules, while above the cloud points, PNiPAAm precipitates with encapsulated molecules and releases them slowly, producing a long-term depot effect [4,45].

A balance between hydrophilic and hydrophobic properties has also been studied in the antigen-specific immune response elicited by injecting various amide-containing polymers and CII. Four various amines containing polymers polyacrylamide (PAAm), poly-*N*-isopropyl acrylamide (PNiPAAm), poly-*N*-isopropylacrylamide-*co*-poly-*N*-tertbutylacrylamide (PNiPAAm-*co*-PNtBAAm), and poly-*N*-tertbutylacrylamide (PNtBAAm) were synthesized and tested with CII for the development of collagen-induced arthritis in mice. The hydrophobic character in these polymers was increased by increasing the alkyl group *N*-substitution (Figure 2). PAAm was highly hydrophilic, while PNtBAAm held high hydrophobic characteristics. Upon injecting these polymers, PNiPAAm mixed with CII induced arthritis symptoms in 75% of the mice compared to those injected with other polymers plus CII. A balance of both properties might recruit more APCs at the injection site for a high cellular uptake of antigen, which can lead to a more robust response than the highly hydrophilic or hydrophobic polymers [5]. Amide chemistry has proven to be an effective adjuvant to activate an auto-antigen-specific immune response. However, their adjuvant properties need to be addressed with microbial antigens to explore them in vaccines.

## 5. Acid and Ester Chemistry

Polyglycolic acid (PGA) and its copolymers like poly-lactide-*co*-glycolide (PLGA) are familiar examples of the role of amide chemistry in developing antigen-specific immune responses. Generally, PLGA-based polymers are polyesters of lactic and glycolic acids at a molar ratio of 50:50. The molecular weight and copolymer composition influence the PLGA polymers’ immunogenicity via changes in degradation and antigen release profiles [46]. PLGA polymers are highly biocompatible and biodegradable [47]. Under in-vivo conditions, PLGA microparticles are phagocytosed by the phagocytic cells, and hydrolysis occurs inside the specialized vacuoles of these phagocytic cells. The ester linkage of PLGA breaks down into lactic and glycolic acid moieties [46]. This produces an immunosuppressive effect via inhibitory phenotype and resistance maturation of murine bone marrow-derived dendritic cells [48]. The hydrophobicity and release rate of incorporated materials can be varied by changing the ratio of individual lactic and glycolic acid monomers [49]. The presence of higher-level glycolic acid in the polymer results in a rapid burst and release of antigens [50].

On the other hand, the capacity of PLGA microparticles loaded with autoantigenic peptides has reduced the hyperglycemic condition in autoimmune type 1 diabetes in a mouse model [51]. Therefore, PLGA effects were immunosuppressive, and different studies can show activation properties. In another study, the effect of PLGA particles was studied with double-stranded RNA adjuvant Riboxxim and an antigen in anticancer immunotherapy. Antigen and adjuvant-loaded PLGA particles activated the murine and human dendritic cells and upregulated the tumor-specific CD8+ T cell responses. This PLGA-based formulation inhibited tumor growth, prevented metastasis, and increased the survival of the mice possessing tumor load [24].

In addition to the amide linkage, the opposite charge effect is also studied using PLGA particles. For instance, three different kinds of PLGA particles with additional charges were synthesized in a study to observe their impact on the immune system. The immune potential of an antigen-loaded model negatively charged with PLGA NPs Angelica sinensis polysaccharide (ASI-PLGA-Ova), positively charged with polyethyleneimine PLGA NPs (PEI-PLGA-Ova), and PEI-modified negatively charged ASI-PLGA-Ova particles was demonstrated in mice. Unlike negatively charged NPs, both PEI-coated PLGA NPs were biased for antigen escape from the endosome, leading to the antigen delivery into the cytoplasm to enhance cross-presentation. Moreover, both PEI-modified NPs induced more effective long-term antigen-specific immune responses in mice [25]. In a different study, another acid moiety containing polymer polymethylmethacrylate (PMMA)-based NPs was demonstrated to deliver the HIV microbial Tat antigen to activate the antigen-specific immune system. NPs constituted PMMA in the inner core while positively charged PEG molecules reside on the outer shell for adsorption of microbial antigen. Upon intramuscular injection of plasmid, pCV-tat delivered through PMMA-PEG NPs successfully activated Tat-specific humoral and cellular immune responses. However, the response was biased toward the Th1 cell signaling pathway [26]. PLGA polymer can successfully present antigens to stimulate cellular and humoral immune responses [52]. Dendritic cells and macrophages are involved in the internalization of PLGA particles [53] and release antigens to present via the MHC class I pathway to induce cytotoxic T cells [54]. Apart from inducing secretion of cytokines from dendritic cells [50], the PLGA particles can also induce IgG and IGA responses [50,55,56].

Polyacrylic acid polymers (carbomers) are used as adjuvants in veterinary vaccines. The original synthetic carbomer, Carbopol, is an anionic polymer of acrylic acid cross-linked with polyalkenyl ethers or divinyl alcohol and used in humans for topical application and drug delivery purposes. Antigens can be directly mixed with carbopol gel, which has no apparent toxic effects in animals [57,58]. Carbopol promotes cellular immunity without PRR activation [59].

## 6. Alcohol Chemistry

In general, alcohol functionality in a polymer structure contributes a hydrophilic property and inhibits the binding of proteins under in-vivo conditions. Polyethylene glycol (PEG) is the most common example of a polymer used in biomedical applications. Hypersensitive reactions, including life-threatening anaphylaxis, are the most common side reactions documented in preclinical and clinical studies. For instance, anti-PEG IgM antibodies were observed in mice vaccinated with PEGylated liposomes containing oligonucleotides as an antigen. Furthermore, immunized mice developed anaphylactic shock after the second dose of PEGylated liposomes [27]. In a different study, PEGylated liposomes encapsulated with DNA induced PEG-specific antibody responses in mice. This led to common hypersensitive reactions including lethargy, puffiness around the eyes, cyanosis, labored breathing, and mortality [60]. Therefore, in-vivo performance of other alcohol-based polymers needs to be demonstrated in the future.

## 7. Ionic Interactions

Ionic interactions are vital in activating immune responses by stabilizing loaded antigens. For instance, the cationic trimethyl chitosan polymer interacts with a peptide antigen which is further coupled with polyglutamic acid (PGA). The peptide antigen contains a B cell conserved sequence and a universal T cell epitope conjugated PGA via cycloaddition reaction, producing the anionic conjugate. The cationic conjugate further ionically interacts with cationic TMC to form NPs. These NPs induced highly systemic and mucosal immune responses like the antigen mixed with the cholera toxin B subunit, which served as a positive control. The systemic antibodies were observed to be opsonized against the group A streptococcus pathogenic strains [61]. In this sense, the effect of a quaternization charge on the activation of the immune system has also been studied. For example, 2-hydroxypropyl trimethyl ammonium chloride chitosan (HTCC) hydrogel with different degrees of quaternization (0, 21, 41, 60, 80%) was synthesized. The positive charge increased as the DQ increased, increasing the hydrogel’s gelation time. Upon nasal administration of this different hydrogel with the H5N1 vaccine, it was found that 41% quaternization HTCC hydrogel induced a more potent H5N1-specific systemic immune response. HTCC with 0% quaternization caused a high level of mucosal immune responses, compared to the other hydrogels, because of more interactions with the mucosal surface [28] (Figure 3). In addition to protein-based antigens, nucleic acid antigens have been explored with polymers with ionic charges. Polyethyleneimines (PEIs) are the common examples which have been constantly explored in studies on the delivery of NA-based vaccines. PEI-based polymers form nanocomplexes upon mixing with NA-based antigen through electrostatic interactions. These nano complexes can be easily internalized by APCs for delivery of NAs. The immune response generated in NA-based studies is superior to that of the NA alone [29,30].

## 8. Coordination Interactions

The coordination interactions between metal ions and organic ligands attract biomedicine scientists to apply them for immunomodulation of the immune system. For instance, Imidazole-based compounds provide a better scaffold for developing therapeutic drugs [62]. A coordination complex of Zn ions with 4-phenyl imidazole (PI) was explored as an inhibitor of indoleamine 2,3-dioxygenase in cancer immunotherapy. The Zn-PI complexes were modified with PEG and loaded with a cancer drug, doxorubicin, to target the cancer cells. The Zn ion of the complex influenced the calreticulin proteins on the cancerous cells to enhance DOX-triggered immunogenic cell death. However, Zn ligand PI helped reverse the immunosuppressive environment by inhibiting the IDO enzyme. The DOX-PEG-loaded Zn-PI complexes effectively inhibited tumor growth via chemo- and immune-therapy ways [31]. In a different study, the immune potential of Au-phosphane dithiocarbamate coordination complexes was demonstrated using ovarian cancer cells. Le et al. synthesized a series of Au-dialkyl dithiocarbamate complexes that were an excellent cytotoxic for cancer cells by inducing oxidation stress, endoplasmic reticulum stress-mediated oxidation stress, and endoplasmic reticulum stress-mediated p53-independent apoptosis of cancer cells. Moreover, these complexes interact with the cancer cells’ calreticulin proteins to activate anticancer immune responses. However, the true potential of these Au-based complexes needs to be addressed in the future [63].

## 9. Combination of Different Interactions

The combination of different functionalities has also been demonstrated to activate an antigen-specific immune response. For instance, the role of different interactions was studied with gold NPs via attaching different functional groups on NPs’ surface. Despite its ease in conjugation, Au displays an excellent immunomodulation property [64]. Different Au NPs with other hydrophobic characteristics were synthesized, and their capacity to activate the immune system was demonstrated. The Au NPs with R groups possessing a more cyclic structure were found to be more potent in immune system activation. The explanation for this observation was their higher hydrophobic index compared to the other Au NPs with hydrophilic R groups. The R groups of alcohol and primary amino groups responded poorly, as a low cytokine expression level was found after exposing them to murine splenocytes. Upon administration to mice, a similar effect of NPs’ hydrophobicity was observed [65].

## 10. Conclusions

Decoding polymer interactions with the biological system is needed to improve or develop new polymer systems as adjuvants. In our current knowledge, polymers physically or chemically interact with biologics via different functional groups and control their release in vivo. In some cases, polymers act as agonists for innate immune receptors to activate the immune system. Altering polymer functionality has been shown to have a direct biological effect, to some extent. However, more fine tuning is needed to design polymers to activate the immune system more effectively.

## Figures and Tables

**Figure 1 vaccines-11-01395-f001:**
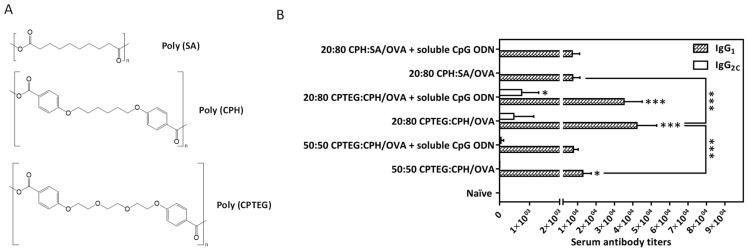
Adjuvant potential of polyanhydrides. (**A**) Chemical structures of different polyanhydrides, (**B**) their adjuvant potential with Ovalbumin (Ova) antigen and CpG adjuvant in mice. Mean values ± SD were plotted in the presented graph. * *p* < 0.05, *** *p* < 0.001. Figure reproduced with permission from [22].

**Figure 2 vaccines-11-01395-f002:**
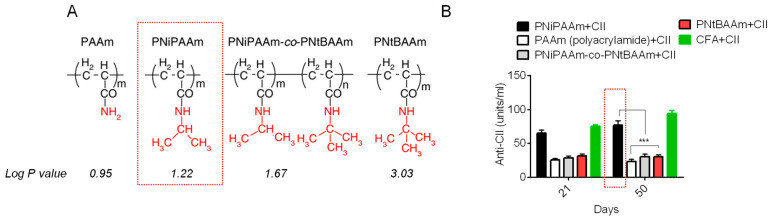
Adjuvant potential of polyamides. (**A**) Chemical structures of amine-containing polymers with increasing hydrophobic properties, (**B**) anti-CII response in mice serum injected with different polymers mixed with CII. Mean values ± SD were plotted in presented graph data are plotted in form of mean, *** *p* < 0.001. figure, reproduced with permission from [5].

**Figure 3 vaccines-11-01395-f003:**
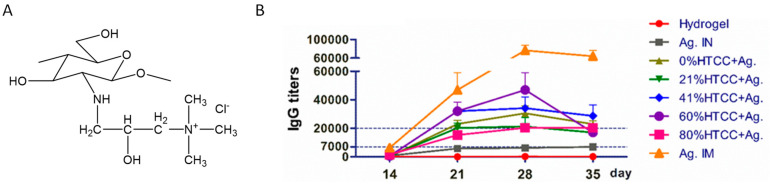
The adjuvant potential of positively charged polymer. (**A**) Chemical structures of modified 2-hydroxypropyl trimethyl ammonium chloride chitosan (HTCC), (**B**) its adjuvant potential in different proportions with H5N1 vaccine in mice. Figure reproduced with permission from [28].

**Table 1 vaccines-11-01395-t001:** Summary of various polymer chemistries utilized for different immunological applications.

Functionality	Polymer	Antigen/Vaccine	Application	Reference
Anhydride 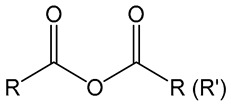	Polyanhydride- copolymers (CPTEG+CPH ^§^, CPTEG+SA ^§§^)	Ovalbumin	Activation of immune system	[22]
Zwitterionic 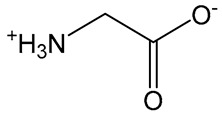	Poly (carboxybetaine methacrylate) (pCBMA)	Immunoglobulin (Ig)	Therapeutic purpose	[23]
Amide 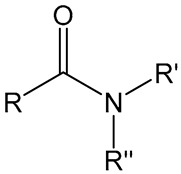	Poly-*N*-isopropyl acrylamide (PNiPAAm),	CII ^†^	Development of experimental autoimmune arthritis mouse model	[5,6,7]
	Polyacrylamide (PAAm),	CII		
	Poly-*N*-isopropylacrylamide-*co*-Poly-*N*-tertbutylacrylamide (PNiPAAm-*co*-PNtBAAm)	CII		
	Poly-*N*-tertbutylacrylamide (PNtBAAm)	CII		
Acid and ester 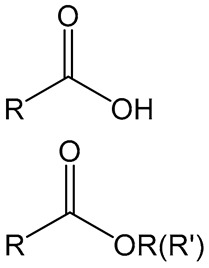	Poly-lactide-*co*-glycolide (PLGA)	Cancer antigen	Anticancer immunotherapy	[24]
	Polyethyleneimine PLGA NPs (PEI-PLGA	Ova	Immune system activation	[25]
	Polymethylmethacrylate (PMMA)	HIV tat antigen	Immune system activation	[26]
Alcohol 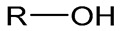	Polyethylene glycol (PEG)	Nucleic acids	Therapy	[27]
Electrostatic interactions 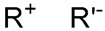	2-hydroxypropyl trimethyl ammonium chloride chitosan (HTCC)	Influenza vaccine	Immune system activation	[28]
	Polyethyleneimines (PEIs)	Nucleic acid antigen	Immune system activation	[29,30]
Coordination interactions 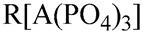	Zn-PI complex (A complex of Zn ions with 4-phenyl imidazole (PI))	Doxorubicin	Anticancer immunotherapy	[31]

^§^ 8-bis-(p-carboxy phenoxy)-3,6-dioxaoctane + 1,6-bis-(p-carboxy phenoxy)-hexane, ^§§^ 8-bis-(p-carboxy phenoxy)-3,6-dioxaoctane + sebacic anhydride, ^†^ collagen type II.

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
