# Peer review of "Polymer Chemistry Defines Adjuvant Properties and Determines the Immune Response against the Antigen or Vaccine"

_vaccines, 2023, doi:10.3390/vaccines11091395_

Round 1

Reviewer 1 Report

Manuscript entitled" Polymer chemistry defines adjuvants properties and determines the immune response against an antigen or a vaccine" by Shakya and Nandakumar is well organized and well written scientific information, I have no objections to accept the manuscript with following condition

1) Title is too long and avoid word adjective or

2) Authors shall give the bonding pattern in each type of chemistry synthesis

3) In the interactive chemistry , authors mentioned about GNPs to R group to the ammonium ions, whats the precise meaning of this statement. How AuNP with R group helps in activation of immune system. I feel surface interaction plays a significant role in the bio activation, Authors requested to cite PMID:35005942. where investigators descriptively narrated the surface chemistry

4) Conclusion is not precise, authors should add further points to this section with some notes on futuristic explanation

Author Response

Manuscript entitled" Polymer chemistry defines adjuvants properties and determines the immune response against an antigen or a vaccine" by Shakya and Nandakumar is well organized and well written scientific information, I have no objections to accept the manuscript with following condition.

Response- Authors are thankful to reviewer for the positive comments, and revised manuscript has taken care all his concerns.

1) Title is too long and avoid word adjective or

Response- The title has been modified now.

2) Authors shall give the bonding pattern in each type of chemistry synthesis.

Response- A summary table including bonding pattern has been included in revised manuscript.

3) In the interactive chemistry , authors mentioned about GNPs to R group to the ammonium ions, whats the precise meaning of this statement. How AuNP with R group helps in activation of immune system. I feel surface interaction plays a significant role in the bio activation, Authors requested to cite PMID:35005942. where investigators descriptively narrated the surface chemistry.

Response- Suggested reference has been cited in the Au section.

4) Conclusion is not precise, authors should add further points to this section with some notes on futuristic explanation.

Response- We have modified conclusion now.

Reviewer 2 Report

Summary: The authors present a review of different polymer chemistries used in adjuvants.  There is a brief introduction of how polymers influence immune responses followed by sections on different chemistries that have been used.  The authors end the review by stating that further knowledge and understanding of how polymers interact with biological systems is essential to improve new therapeutics.

Reviewers’ summary:  The manuscript does a good job of introducing how polymers may influence immune responses and discusses different chemistries by section, to help the reader focus.  The sections start with a description of the chemistry, examples of polymers, brief review of published findings, and a statement on how the polymer influenced the immune response.  The authors do a great job on the chemistry portion with sufficient knowledge and depth but tend to describe immune responses at a superficial level.  I would recommend that the manuscript be revised to improve the depth of discussions on how each polymer chemistry impacts the immune system with more details on how the immune system is impacted or affected.  Included are specific examples by line number and a suggestion to improve.

1.       Abstract: Line 11: “an antigen alone is insufficient to activate the immune system”.  This is not accurate.  There are many, many examples of antigens that are not polymers nor require polymers in adjuvants to generate strong immune responses: ovalbumin, diphtheria toxin, and tetanus toxoid are just some examples that have been used for many decades that do not require adjuvants to generate robust immune responses.

2.       Abstract line 12: “assistance in the form of polymers is needed to control the release of antigens under in vivo conditions… to active the immune system efficiently”.  This is not an accurate statement.  Robust immune responses have been generated without the use of polymers, adjuvants, or formulations that control the release of the antigen when presented to a host.

3.       Line 35 and 43: “polymers assist in antigen processing better to activate the immune system” and “adjuvant helps antigens presented to the antigen-presenting cells and enhances costimulatory signals for activation of Th cells”.  What is the mechanism for polymers improving antigen processing and presentation?  Antigen processing is a very complex process that involves enzymatic proteases to generate peptides which are then presented on the cell surface in the context of MHC class I or MHC class II molecules.  These are very different antigen processing pathways, do polymers improve both?  What costimulation signals are being enhanced?  Line 91 discusses how zwitterion polymers coating antigens (enzyme conjugates) suppress the expression of CD40 and CD80 (classical costimulatory molecule) on APCs.  Suggesting that polymers suppress costimulation, not enhance them.  Thus, the authors might adjust the description of polymers to include their ability to modulate costimulatiory signals.

4.       Line 46: “mechanistically, no specific cell signaling pathways are known in polymer mediated activation of the immune system”.  I do not think this is an accurate statement.  Recommend the authors review the biology associated with T cell independent antigens-type II.  These are categorized as polysaccharides, glycolipids, nucleic acids, etc. (e.g., polymers) that bind directly to the antibody receptors on B cells and stimulate intracellular signaling pathways (Syk, PI3K, PLC gamma, PDK1, etc.) that lead to production of antibodies without engaging T cells. These immune responses are measurable and provide protection.

5.       Line 66: the authors discuss how polymers may promote opsonization of particles which may be attributed to danger signals.  Opsonization is usually described in terms of complement factors or antibodies binding to antigens that help mediate recognition by APCs.  Polymeric antigens are known to be very efficient with this process. These are mediated by complement receptors or Fc receptors on the APCs.  However, the authors attribute this mechanism to danger signals.  Can the authors expand on what danger signals are and how they facilitate and generate the immune response?  Is there a reference for this?

6.       Line 88: The authors describe a publication by Li et al (ref 8) where a zwitterionic polymer used in a uricase conjugate enhanced the PK profile while inhibiting the immunogenicity properties of the conjugate.  Specifically, it affected APCs via suppression of CD40 and CD80.  Figure 2 and reference 8 are provided for the reader for reference.  However, figure 2 describes amine containing polymers and anti-CII (collagen type II) specific immune responses.  Figure 2 does not show the ZPS uricase conjugate (line 88). Polymers such as PEG, are commonly used in protein based therapeutics to improve PK half-life via increased drug stability and efficacy.  Is it possible that ZPS improved molecule stability and thus the PK profile?  How would a polymer inhibit immunogenicity properties?  Is that via masking of epitopes?

7.       Lines 118-122: The section describes the ability of polymers to recruit APCs to the injection site thus increasing the cellular update of antigen.  What is the proposed mechanism by which these polymers attract APCs?  Is this via stimulation of innate immunity?

8.       Lines 133-137: This section describes how PLGA particles may have immunosuppressive effects on the maturation of dendritic cells.  Suggesting that polymers can suppress immune responses as opposed to enhancing the immune responses as discussed in the introduction.

9.       Lines 153-156: There is a description of how PEI-coated PLGA NPs biased (??) for antigen escape from the endosome, leading to cross-presentation?  Are the authors describing the mechanism by which cytoplasmic antigens get processed and presented via MHC class I presentation pathway?

10.   Line 164: Alcohol chemistry section.  PEG is commonly used by pharmaceutical companies to stabilize biologics leading to improved half-life and better efficacy.  However, there are instances where anti-PEG immune responses are observed in clinical trial subjects, which is a concern by regulatory agencies.  It’s not uncommon for the FDA to ask for an assessment of anti-PEG immune responses in clinical trial subjects in the drug approval process. Perhaps the authors would include this topic in this section which would further show how polymers can impact human immunological reactions.

11.   Suggest a table be included in the manuscript that shows the different polymer chemistries, key characteristics, and immune modulation properties.  This would help highlight the key points of each chemistry and help the reader compare and contrast the different polymer categories. 

No issues or concerns with the quality of the English. 

Author Response

Summary: The authors present a review of different polymer chemistries used in adjuvants.  There is a brief introduction of how polymers influence immune responses followed by sections on different chemistries that have been used.  The authors end the review by stating that further knowledge and understanding of how polymers interact with biological systems is essential to improve new therapeutics.

Reviewers’ summary:  The manuscript does a good job of introducing how polymers may influence immune responses and discusses different chemistries by section, to help the reader focus.  The sections start with a description of the chemistry, examples of polymers, brief review of published findings, and a statement on how the polymer influenced the immune response.  The authors do a great job on the chemistry portion with sufficient knowledge and depth but tend to describe immune responses at a superficial level.  I would recommend that the manuscript be revised to improve the depth of discussions on how each polymer chemistry impacts the immune system with more details on how the immune system is impacted or affected.  Included are specific examples by line number and a suggestion to improve.

Response- Authors are thankful to reviewer for giving the positive and constructive comments to revise our manuscript. We have addressed all reviewers’ concerns in the revised manuscript.

  1. Abstract: Line 11: “an antigen alone is insufficient to activate the immune system”.  This is not accurate.  There are many, many examples of antigens that are not polymers nor require polymers in adjuvants to generate strong immune responses: ovalbumin, diphtheria toxin, and tetanus toxoid are just some examples that have been used for many decades that do not require adjuvants to generate robust immune responses.

Response- We agree with the reviewer’s point. The sentence has been corrected now.

  1. Abstract line 12: “assistance in the form of polymers is needed to control the release of antigens under in vivo conditions… to active the immune system efficiently”.  This is not an accurate statement.  Robust immune responses have been generated without the use of polymers, adjuvants, or formulations that control the release of the antigen when presented to a host.

Response- The statement has been corrected now.

  1. Line 35 and 43: “polymers assist in antigen processing better to activate the immune system” and “adjuvant helps antigens presented to the antigen-presenting cells and enhances costimulatory signals for activation of Th cells”.  What is the mechanism for polymers improving antigen processing and presentation?  Antigen processing is a very complex process that involves enzymatic proteases to generate peptides which are then presented on the cell surface in the context of MHC class I or MHC class II molecules.  These are very different antigen processing pathways, do polymers improve both?  What costimulation signals are being enhanced?  Line 91 discusses how zwitterion polymers coating antigens (enzyme conjugates) suppress the expression of CD40 and CD80 (classical costimulatory molecule) on APCs.  Suggesting that polymers suppress costimulation, not enhance them.  Thus, the authors might adjust the description of polymers to include their ability to modulate costimulatiory signals.

Response- Authors have changed the sentence to make better sense about the polymer. Yes, reviewer is right, till date there is no study available which can prove the polymer directly have effect on antigen processing or presentation. However, polymers can indirectly influence the antigen processing process via regulating release of an antigen (doi:10.1016/j.actbio.2011.03.023). Moreover, polymers do affect the costimulatory signals as Tamayo et al., has demonstrated the polyanhydide NPs capacity to enhance the costimulatory signals (https://journals.asm.org/doi/10.1128/CVI.00164-10).

  1. Line 46: “mechanistically, no specific cell signaling pathways are known in polymer mediated activation of the immune system”.  I do not think this is an accurate statement.  Recommend the authors review the biology associated with T cell independent antigens-type II.  These are categorized as polysaccharides, glycolipids, nucleic acids, etc. (e.g., polymers) that bind directly to the antibody receptors on B cells and stimulate intracellular signaling pathways (Syk, PI3K, PLC gamma, PDK1, etc.) that lead to production of antibodies without engaging T cells. These immune responses are measurable and provide protection.

Response- Thanks for your comments. I think, reviewer is talking about natural polymers like bacterial polysaccharides who are known to activate the intracellular signaling pathways. In the current manuscript, we are majorly focusing on immunomodulation properties of synthetic polymers.

  1. Line 66: the authors discuss how polymers may promote opsonization of particles which may be attributed to danger signals.  Opsonization is usually described in terms of complement factors or antibodies binding to antigens that help mediate recognition by APCs.  Polymeric antigens are known to be very efficient with this process. These are mediated by complement receptors or Fc receptors on the APCs.  However, the authors attribute this mechanism to danger signals.  Can the authors expand on what danger signals are and how they facilitate and generate the immune response?  Is there a reference for this?

Response- The sentence has been further expanded along with new references.

  1. Line 88: The authors describe a publication by Li et al (ref 8) where a zwitterionic polymer used in a uricase conjugate enhanced the PK profile while inhibiting the immunogenicity properties of the conjugate.  Specifically, it affected APCs via suppression of CD40 and CD80.  Figure 2 and reference 8 are provided for the reader for reference.  However, figure 2 describes amine containing polymers and anti-CII (collagen type II) specific immune responses.  Figure 2 does not show the ZPS uricase conjugate (line 88). Polymers such as PEG, are commonly used in protein-based therapeutics to improve PK half-life via increased drug stability and efficacy.  Is it possible that ZPS improved molecule stability and thus the PK profile?  How would a polymer inhibit immunogenicity properties?  Is that via masking of epitopes?

Response- Relevant information has been added in the respective sections. Please follow the changes in red font.

  1. 7.Lines 118-122: The section describes the ability of polymers to recruit APCs to the injection site thus increasing the cellular update of antigen.  What is the proposed mechanism by which these polymers attract APCs?  Is this via stimulation of innate immunity?

Response- Yes, there is possibility that polymers may act as agonist for TLRs reside on the cells to activate the immune system. However, in general, it has been found that their role more on side of slow or sustain release of the antigen to keep activating the immune system.

  1. Lines 133-137: This section describes how PLGA particles may have immunosuppressive effects on the maturation of dendritic cells.  Suggesting that polymers can suppress immune responses as opposed to enhancing the immune responses as discussed in the introduction.

Response- Yes, authors agree with reviewer. The polymers can act as an adjuvant to enhance or suppress the immune response. The type of response polymers produce depends on majorly chemistry and other factors like format and molecular weight of the polymer.

  1. Lines 153-156: There is a description of how PEI-coated PLGA NPs biased (??) for antigen escape from the endosome, leading to cross-presentation?  Are the authors describing the mechanism by which cytoplasmic antigens get processed and presented via MHC class I presentation pathway?

       Response- Generally, polyanhydrides like PLGA are unbiased for the immune response. They assist a particular antigen via protecting it from in-vivo condition. We do not know exactly how the polymer helps the antigen escape from the phagocytosis.

  1. Line 164: Alcohol chemistry section.  PEG is commonly used by pharmaceutical companies to stabilize biologics leading to improved half-life and better efficacy.  However, there are instances where anti-PEG immune responses are observed in clinical trial subjects, which is a concern by regulatory agencies.  It’s not uncommon for the FDA to ask for an assessment of anti-PEG immune responses in clinical trial subjects in the drug approval process. Perhaps the authors would include this topic in this section which would further show how polymers can impact human immunological reactions.

      Response- We agree with reviewer. However, we do not want t to emphasize so much on side reactions notice due to polymer like PEG. The current article emphasizes more on different polymer chemistries and their impact on immune system.

  1. Suggest a table be included in the manuscript that shows the different polymer chemistries, key characteristics, and immune modulation properties.  This would help highlight the key points of each chemistry and help the reader compare and contrast the different polymer categories.

Response- A summary table showing different polymer chemistry and their immunological applications has been included now.

Reviewer 3 Report

General Comment: The authors do not have enough information about polymer chemistry and its immunological properties, as described in the title. Also, this review does not focus intensely on the influence of polymer chemistry and vaccine development. In addition, the sections are not well linked and do not have an intro about the described polymers. 

Specific comments about the body of the review

Title (see the general comment)

Abstract (comments)

Antigens alone are insufficient to activate the immune system. Superantigens and bacterial toxins are good examples of potent antigens. 

It is unclear if the polymers play an essential function when used with a vaccine delivery platform or directly to protect the antigen.

Introduction (comments)

The authors do not have a linking paragraph between the introduction and the coming sections in the review. It makes it difficult to understand the sections after the intro.

polymers act as an adjuvant to improve the efficacy of an antigen (Line 42). 

The line does not have reference. 

B cells to produce the antibodies (Line 45)

The terminally differentiated B cell (Plasma cells) produce antibodies.

activation of complement pathways, and interactions with pattern recognition receptors like toll-like or C-type lectin receptor pathways are likely to be the possible mechanisms (Line 47-49)

The authors do not clarify if the Polymer modulates innate or adaptive immunity; initially, they refer to cellular immunity and then innate immunity without information that can link the immunological branches.

Section: Anhydride chemistry (comments)

The authors do not introduce the anhydride chemistry fundaments and the importance of the ratio from different chemical components.

It is unclear if the source of Figure 1 and its information is also unclear.

As discussed previously, it is not clear if this type of polymer is involved in the vaccine platform incorporation (APC) or with the antigen. 

This section does not have enough references. 

Section: Zwitterionic chemistry (comments)

This section has limited information; I recommend expanding it to focus more on the MOA. 

The authors do not clarify the properties of the antigen to be a good candidate for polymers based on the Zwitterionic chemistry.

Minor revision

Author Response

General Comment: The authors do not have enough information about polymer chemistry and its immunological properties, as described in the title. Also, this review does not focus intensely on the influence of polymer chemistry and vaccine development. In addition, the sections are not well linked and do not have an intro about the described polymers. 

Specific comments about the body of the review

Title (see the general comment)

1) Abstract (comments)

Antigens alone are insufficient to activate the immune system. Superantigens and bacterial toxins are good examples of potent antigens. 

It is unclear if the polymers play an essential function when used with a vaccine delivery platform or directly to protect the antigen.

Response- We are agreeing with reviewer. The abstract has been modified now.

2) Introduction (comments)

The authors do not have a linking paragraph between the introduction and the coming sections in the review. It makes it difficult to understand the sections after the intro.

Response- The link sentences have been incorporated in manuscript to get a better flow.

3) polymers act as an adjuvant to improve the efficacy of an antigen (Line 42). 

The line does not have reference. 

Response- The references have been added now.

4) B cells to produce the antibodies (Line 45)

The terminally differentiated B cell (Plasma cells) produce antibodies.

Response- The sentence has been corrected now. 

5) Activation of complement pathways, and interactions with pattern recognition receptors like toll-like or C-type lectin receptor pathways are likely to be the possible mechanisms (Line 47-49)

The authors do not clarify if the Polymer modulates innate or adaptive immunity; initially, they refer to cellular immunity and then innate immunity without information that can link the immunological branches.

Response- In one hand, some polymer as an adjuvant can assist the antigen to activation of humoral and cellular immune response (PMID-21543351). On the other hand, other kind of polymers like polyanhydrides can acta as agonist for some TKRs to activate the innate immunity (PMID- 20631332).

6) Section: Anhydride chemistry (comments)

The authors do not introduce the anhydride chemistry fundaments and the importance of the ratio from different chemical components.

It is unclear if the source of Figure 1 and its information is also unclear.

As discussed previously, it is not clear if this type of polymer is involved in the vaccine platform incorporation (APC) or with the antigen. 

This section does not have enough references. 

Response- The relevant details of polyanhydrides has been added to this section to give a better opinion about the polyanhydrides.

7) Section: Zwitterionic chemistry (comments)

This section has limited information; I recommend expanding it to focus more on the MOA. 

The authors do not clarify the properties of the antigen to be a good candidate for polymers based on the Zwitterionic chemistry.

Response- Few more examples has been added in this section to discuss more zwitterionic polymers.

Round 2

Reviewer 2 Report

The authors have made significant changes to the manuscript which have greatly improved it.  This is a very well written review of the field and should be a great resource for scientists.  The reviewer would like to thank the authors for their patience and for the improvements.  Very nice!

Author Response

Thank you for your comments.